# A platform-independent framework for phenotyping of multiplex tissue imaging data

**Mansooreh Ahmadian**[1], **Christian Rickert**[2], **Angela Minic**[2], **Julia Wrobel**[1], **Benjamin G. Bitler**[3], **Fuyong Xing**[1], **Michael Angelo**[4], **Elena W. Y. Hsieh**[2,5], **Debashis Ghosh**[1], **Kimberly R. Jordan**[2]*

**1** Department of Biostatistics and Informatics, Colorado School of Public Health, University of Colorado Anschutz Medical Campus, Aurora, Colorado, United States of America, **2** Department of Immunology and Microbiology, University of Colorado Anschutz Medical Campus, Aurora, Colorado, United States of America, **3** Division of Reproductive Sciences, Department of OB/GYN, University of Colorado Anschutz Medical Campus, Aurora, Colorado, United States of America, **4** Department of Pathology, Stanford University, Stanford, California, United States of America, **5** Pediatrics, Section of Allergy and Immunology, University of Colorado Anschutz Medical Campus, Aurora, Colorado, United States of America

* kimberly.jordan@cuanschutz.edu

**Data Availability Statement:** The authors confirm that all data underlying the findings are fully available without restriction. This work describes two distinct datasets: breast cancer and ovarian

## Abstract

Multiplex imaging is a powerful tool to analyze the structural and functional states of cells in their morphological and pathological contexts. However, hypothesis testing with multiplex imaging data is a challenging task due to the extent and complexity of the information obtained. Various computational pipelines have been developed and validated to extract knowledge from specific imaging platforms. A common problem with customized pipelines is their reduced applicability across different imaging platforms: Every multiplex imaging technique exhibits platform-specific characteristics in terms of signal-to-noise ratio and acquisition artifacts that need to be accounted for to yield reliable and reproducible results. We propose a pixel classifier-based image preprocessing step that aims to minimize platform-dependency for all multiplex image analysis pipelines. Signal detection and noise reduction as well as artifact removal can be posed as a pixel classification problem in which all pixels in multiplex images can be assigned to two general classes of either I) signal of interest or II) artifacts and noise. The resulting feature representation maps contain pixel-scale representations of the input data, but exhibit significantly increased signal-to-noise ratios with normalized pixel values as output data. We demonstrate the validity of our proposed image preprocessing approach by comparing the results of two well-accepted and widely-used image analysis pipelines.

## Author summary

Multiplex tissue imaging techniques are powerful tools increasingly used to characterize the structural and functional states of cells in situ. Each multiplex imaging platform exhibits unique characteristics such as signal-to-noise ratio and artifacts that need to be accounted for before to accurately analyze the data. Here, we present an image preprocessing framework that removes noise, artifacts, and platform-specific characteristics of the

cancer data. The breast cancer dataset, previously studied in Keren et al. [https://doi.org/10.1016/j.cell.2018.08.039], is publicly accessible at https://mibi-share.ionpath.com. Comprehensive information, including channel images, segmentation masks, and cell identities for the ovarian cancer data, can be obtained from https://github.com/himsr-lab/CU-PhenoNorm. The analysis code can also be found in the same GitHub repository.

**Funding:** KRJ was supported by NIH (P30CA046934). BGB supported by DOD, Award OC170228 and the Kay L. Dunton Endowed Memorial Professorship In Ovarian Cancer Research. MA supported by NIH/NCATS Colorado CTSA Grant Number UL1 TR002535. EWYH was supported by the Lupus Research Alliance, Lupus Innovation Award. The funding agencies had no role in study design, data collection and analysis, decision to publish, or preparation of the manuscript.

**Competing interests:** The authors have declared that no competing interests exist.

images and generates normalized, high-quality, and reproducible data for subsequent stages of analyses. Our platform-independent solution performs multiple tasks of denoising, artifact correction, and normalization using a single pixel classification step, eliminates the need for any further normalization process, and works across all tested multiplex imaging technologies.

## Introduction

Multiplex tissue imaging technologies such as Multiplexed Ion Beam Imaging (MIBI) [1, 2], Imaging Mass Cytometry (IMC) [3], CO-Detection by indEXing (CODEX) [4, 5], Multiplexed Immunofluorescence (MxIF) [6, 7], and cyclic Immunofluorescence (cycIF) [8] provide researchers with a wealth of information on the single cell level that illustrates the complexity and heterogeneity of tissue samples. Comprehensive surveys providing an overview of these technologies are readily available [9, 10]. By preserving the spatial context of >40 markers measured simultaneously on a tissue, these advanced technologies have opened new avenues for biological discoveries in healthy and diseased microenvironments [11–19]. A substantial challenge remains, however, in developing accurate, robust, and automated computational pipelines for the analyses and interpretation of these complex high-dimensional imaging data.

A high-dimensional tissue image consists of a set of antibody-based visualizations of multiple parameters (markers) measured with fluorescence or mass spectrometry readouts. The signal intensity in each image channel is directly proportional to the expression level of the corresponding marker bound to its target. Since expression of specific marker combinations is the key determinant of cellular phenotypes, a primary step in the identification of cellular phenotypes is to accurately quantify signal intensities across channels. Accurate quantification of these signal intensities can be hindered by instrumental noise, acquisition artifacts, or experimental variability that are often incorporated in the imaging data during sample preparation and data acquisition. Therefore, most imaging platforms require denoising and artifact removal techniques prior to accurate quantification of signal intensities [2, 20–23]. Remaining artifacts may have far-reaching consequences in downstream analyses, potentially leading to inaccurate cellular identification and false conclusions in statistical comparisons or spatial analyses. Particularly for high-parameter imaging techniques that rely on unsupervised clustering algorithms for cell-type identification, high-quality images are required for robust and accurate quantitative outputs. The data must also be appropriately normalized to limit tissue and batch variability. Without a proper normalization, sample-to-sample intensity variations can cause cells to cluster by individual samples rather than by cell types [24, 25]. However, choosing an appropriate normalization method is a critical task, as the accuracy of cell-type identification has been shown to depend more on the choice of normalization approach than on the clustering algorithm [26]. Machine learning methods that can extract features less sensitive to intensity are therefore adopted [27]. Ideally, high-quality imaging data should satisfy the following criteria: 1) the data should exhibit high signal-to-noise ratios (SNR) and be as free as possible from artifacts; and 2) be appropriately normalized to remove non-biological signal variability within and across acquisition batches and tissues.

Various image processing and filtering methods can be used for denoising and artifact removal [2, 20–22, 28, 29]. For instance, variants of Gaussian filters are used for noise reduction [30–32]. Binary masks can be generated to specifically remove artifacts that differ from real signal by size and pixel distribution. Such techniques have been used for segmentation of microarray imaging data where binary masks are generated following pixel classification that

separates the signal (spots in a microarray image) from background (noise and artifact) [33, 34]. It has been shown that such techniques can limit data loss in preprocessing of the mass cytometry data [34]. Recently, a five-step computational pipeline was proposed to prepare MIBI data for downstream analysis [2]. Steps in the pipeline include background subtraction, necrosis removal, batch normalization, denoising, and aggregate removal. In the first step, non-specific background is removed by subtracting counts from pixels where background noise is present. To specify those pixels with background noise, a binary mask (which is 1 where background exists and 0 elsewhere) is generated by filtering and thresholding a blank channel that is specifically included in the data acquisition process for this purpose. Both the thresholding parameter and the number of counts subtracted are determined manually based on visual inspection by the user. In the second step, artifacts such as areas of necrosis are removed by estimating another binary mask for the necrotic region using morphological opening and closing [2]. The number of counts subtracted and the thresholding parameters are decided by the user. Batch effects are then removed in the third step using quantile normalization. In the fourth step for denoising, a $k$-nearest-neighbor approach is used to generate a binary mask that separates the noisy pixels based on density. The parameter $k$ and the thresholding parameter that separates the high-density pixels (signal) from the low-density pixels (noise) as well as the subtracting pixel counts are determined by the user. In the last step, the images are first smoothed using a Gaussian kernel and then binarized using Otsu's method [35]. This five-step pipeline was recently wrapped in a graphical user interface called MAUI (Mass based imaging Analysis User Interface) [28].

While MAUI facilitates MIBI image preprocessing, it relies on accurate estimation of binary masks with user-defined thresholding parameters that are prone to bias. Thresholding methods separate two distinct features using a single value as the thresholding parameter. However, given the variabilities in complex multiplex tissue imaging data, a single threshold is unlikely to obtain a parameter configuration that perform well across all the images in a dataset. Additionally, estimating binary masks for each artifact in several sequential steps is labor-intensive and not scalable. Finally, new artifacts may appear as data are acquired in new tissue types or with new technologies, requiring additional steps to be added to the aforementioned five-step pipeline.

Here, we aim to create a simple automated framework that provides a single-step unified solution that works across imaging platforms. Our approach combines denoising and removal of various artifacts into a *single* pixel classification step, outputs a feature representation map (FR map) that eliminates the need for any further normalization process, does not rely on estimation of any thresholding parameters and therefore is more robust, and provides a unified solution across multiplex imaging platforms (both mass spectrometry and fluorescence-based imaging). We validate our proposed approach by comparing our results with two well-accepted and widely-used baselines: a) inForm software, a commercial software package for fluorescence-based multiplex imaging analyses and b) MAUI [28], a publicly available computational pipeline used to analyze MIBI data. All code and data have been made available as a resource for the research community [35].

## Results

### A single-step platform-independent framework for preprocessing of multiplex tissue imaging data

Our proposed framework for cellular phenotyping of multiplex tissue imaging data formulates denoising and artifact removal as a pixel classification problem in which the pixels in an image are classified into two classes (Fig 1):

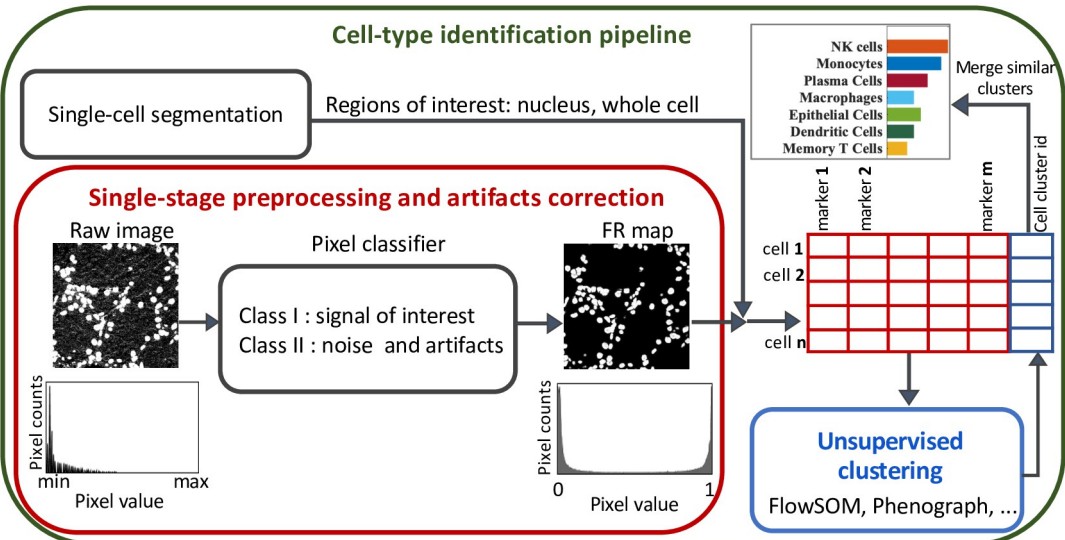

**Fig 1. A general computational pipeline for cellular phenotyping of multiplex tissue imaging data.** We replaced image denoising and preprocessing with our proposed framework as shown in the red box. A pixel classifier is used to classify the pixels in the raw image into two classes: I- desired signal, or II- noise and artifacts. The output of the classifier are two feature representation maps, one for each class, with pixel values between 0 and 1. Marker expression within the border of each cell is then measured from the class I FR maps. The measured single-cell information data (a table with cells in rows and marker expression level in columns) is then used as the input for unsupervised clustering algorithms to identify the cell types.

- class I: pixels with signal of interest (marker signal/positive signal).

- class II: pixels with noise, artifacts, and other platform-specific properties that are not useful for downstream analyses.

The input of the classifier is a training set (labeled data) comprised of user-provided examples for both class I (desired signal) and class II (noise and artifacts) pixels. Features can be extracted by convolving various kernels with the image such as noise reduction filters, edge detectors, and texture kernels. Classifying algorithms, such as Random Forest [36], can then be used to classify each pixel given the extracted features. Several well-established and user-friendly tools have been developed for pixel classification such as Trainable WEKA Segmentation (TWS) [37], QuPath [38], or ilastik [39] and can be used to interactively label training data for signal and noise during image preprocessing. The output of the classifier consists of two FR maps, one for each class. The pixel values in each FR map represent the probability of that pixel belonging to the corresponding class [37]. Pixels with noise and artifacts have either zero or very low values in the FR map of class I, meaning FR maps of class I are free from noise and artifacts. In addition, the values in the FR maps range between 0 to 1. Therefore, the FR maps are normalized and this eliminates the need for further normalization processes.

## Noise and artifact removal from multiplex imaging data

Image noise is generally defined as unwanted random variation that obscures the desired information in an image. Multiplex images are visualizations of bound antibodies, detected either by heavy metal ions conjugated to antibodies in mass spectrometry-based imaging or by fluorescent particles directly or indirectly associated with antibodies in light microscopy-based imaging. Thus, noise in multiplex imaging can be defined as any signal that differs from the

biological structures to which antibodies are specifically bound [28]; and it can be classified into several categories: 1) channel crosstalk, 2) nonspecific antibody staining, and 3) aggregates.

Channel crosstalk or cross-channel contamination is the variable presence of signal from a contaminating channel in a target channel. For fluorescence microscopy imaging techniques, channel crosstalk is caused by the wide and often overlapping emissions spectra of fluoro-chromes. For mass-spectrometry imaging techniques, impurities in the heavy metal ion source used for antibody labeling, modification of the heavy metal ions (by hydrogenation, oxygen-ation, hydroxylation, etc), or for MIBI specifically, gold ions from the slide surface can intro-duce channel crosstalk to the image. For example, a low intensity contaminating signal that mirrors the structure of the real higher intensity signal in the hepatocyte antigen-145 channel (Fig 2A-top) is visible in CD20–161 channel (Fig 2A-middle). To correct this cross-talk, a few examples of the contaminating hepatocyte and real CD20 signals were used to train the pixel classifier and generate an FR map for the positive signal (Fig 2-bottom), in which the contami-nating hepatocyte signal is removed. As demonstrated in the histograms (Fig 2A), the SNR is significantly enhanced in the FR map (green box) compared to the raw image (red box). That is, the values of pixels with positive signal in the FR map group to the far right of the histogram, whereas the rest of the pixels, including those with cross-talk contamination, group to the far left of the histogram and thus are cleanly separated. Another example of cross-talk contamina-tion is gold background (Fig 2B) where signal from the exposed part of the gold slide (contam-inating channel) is observed in the target channel, here Ki-67 (Fig 2B-left). Our approach allows easy removal of this artifact by training the pixel classifier with a few examples of con-taminating pixels (pixels with low intensity that correlate with the bare gold slide) as well as the real Ki-67 signal. The contaminating artifact is completely removed in the resulting FR map (Fig 2B-right) and the Ki-67 signal is clearly separated from the noise pixels, as shown in the histograms.

Non-specific antibody binding is caused by cross-reactivity of antibodies, tissue features that non-specifically bind to antibodies, or instrumental noise. This artifact can appear as either random signal or as dim patterns superimposed on top of biological structures and may have correlation with the histological structure of the tissue [28]. Regardless of the source, the intensity and spatial distribution of this artifact is different from the true positive signal. There-fore, regions with signal from non-specific antibody binding can be easily labeled as undesired signal (class II) using our approach. An example of non-specific antibody staining is necrosis where a necrotic tissue region that exhibits non-specific signal in many channels appears in the pan-cytokeratin channel of a breast cancer tissue (Fig 2C-left). After pixel classification, the artifact is removed in the resulting FR map (Fig 2C-right) and the signal to noise ratio is enhanced, as shown in the histograms.

Aggregates are small high-intensity specks of signal caused by aggregation of antibodies, secondary reagents, or fluorochromes used in the staining process. Due to the concentration of high-intensity pixels, aggregates can be falsely interpreted as positive signal. Fig 2D shows an example of CD163 staining surrounded by noise and aggregates (left). To compare the impact of signal caused by aggregates, pseudo-cells of the same area were drawn by hand (610 pixels) and the mean expression of CD163 was calculated. While area A (m = 1.18) is clearly a CD163+ cell, the mean signal for area B (m = 0.43), which is likely a CD163+ cell, is difficult to distinguish from area C (m = 0.38) that contains positive signal due to an aggregate in the raw image (left). However, after removing noise and aggregates (Fig 2D-right), the mean signal for area C is greatly reduced (m = 0.02) while the mean signal for area B remains high (m = 0.34). Increased SNR after noise and aggregate removal is also demonstrated in the overlaid histo-grams, where the pixel intensities for each pseudo-cell are plotted for the raw image (top) and

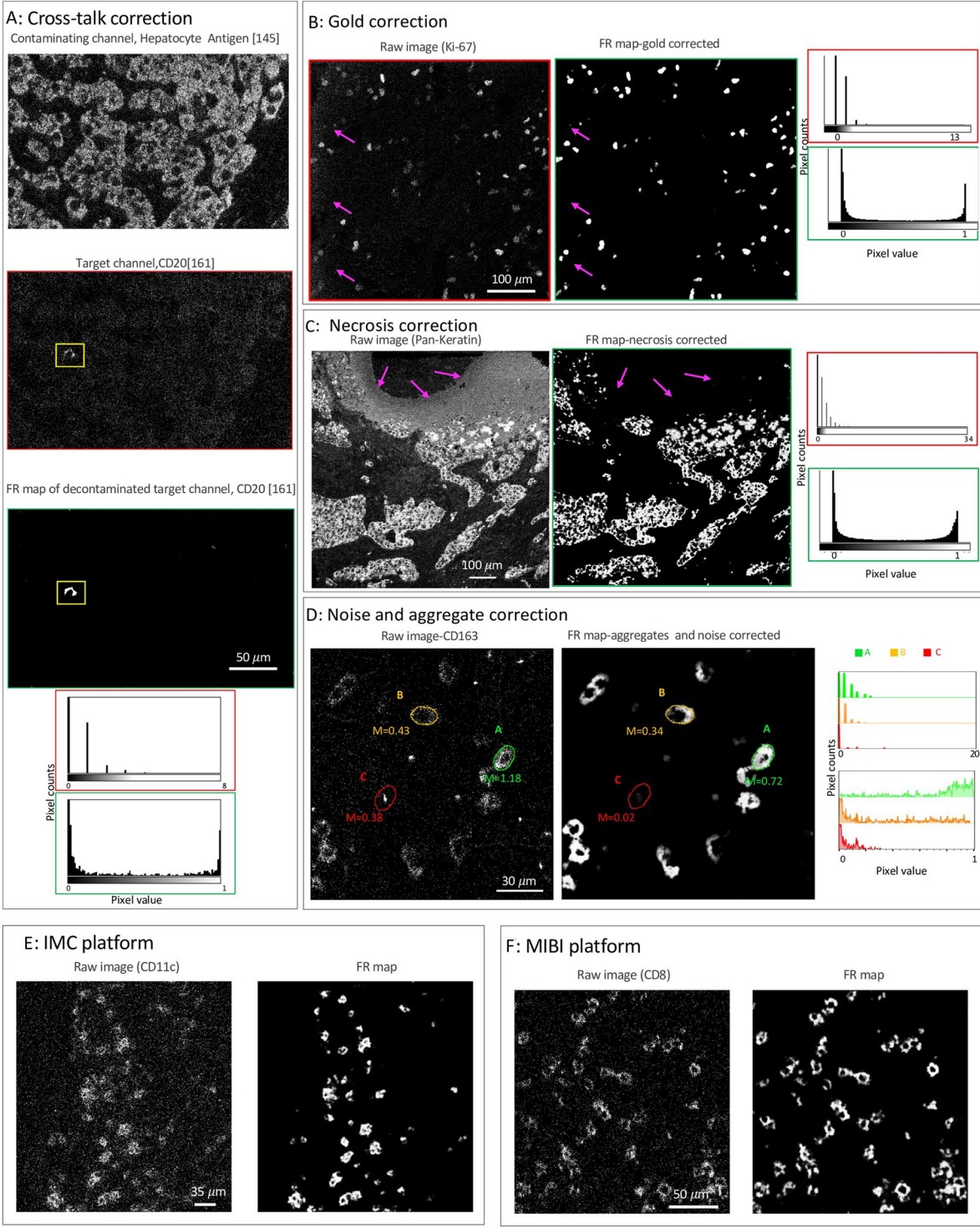

**Fig 2. Removal of various artifacts and noise from MIBI data. A**: An example of cross-channel contamination where a contaminating signal from hepatocyte antigen channel, the oxide of the metal Neodymium (145 m/z + 16 m/z, top), contaminates a target channel, CD20 with Dysprosium (161 m/z, middle). The contamination is removed in the FR map (bottom). The histograms of pixel values for the insets (highlighted in yellow) of the images are displayed both before and after artifacts correction. **B**: Section of Ki-67 marker from ovarian cancer tissue before (left) and after(right) gold removal (a MIBI platform artifact). **C**: Section of breast tissue stained with pan-cytokeratin antibodies before (left) and after (right) removal of necrotic tissue regions. Histograms of pixel values are included for all images (A-C) before and after artifacts correction. **D**: Section of ovarian cancer tissue stained with CD163 antibody before (left) and after noise and aggregates (right) correction. Corresponding histograms of pixel values for the pseudo-cells outlined in green, orange, and red are plotted

before and after noise and aggregate correction. **E:** CD11c staining of lung tissue by IMC (left) and the corresponding FR map (right). **F:** CD8 staining of ovarian cancer tissue by MIBI (left) and the corresponding FR map (right). One classifier is trained for each of the individual raw images shown in panels A-F.

the FR map (bottom). Therefore, our approach can prevent identification of false positive cells from aggregate signal.

Mass based technologies generate pixelated imaging data with low SNR compared to fluorescence-based technologies. Another important advantage of our framework is that the sparse and pixelated signal of Mass-based imaging data (Fig 2D-left) is converted to a continuous signal with high SNR (Fig 2D-right). This property makes the processing of the such data a challenging task because the pixel intensity alone does not carry sufficient information to separate positive signal from noise. The pixel density should be considered along with the intensity [2]. That is, while area A (Fig 2D-left) can be effortlessly called a CD163+ cell given its consistent and high pixel intensity, the signal intensity in area B is much lower and it is the density of the pixels that contributes to forming a cell-like structure. The low intensity signal values of the individual pixels in area B are difficult to distinguish from noise in the raw image (see the corresponding histograms in Fig 2D). However, the spatial information of the pixels contributes to the extracted features in the FR map and thus the classification output. As a result, low-density and low-intensity pixels of noise have very low pixel values in the FR map, while high-density and low-intensity pixels of area B have high pixel values (see the Fig 2D-right and the corresponding histograms). For further illustration, two additional examples of converting mass-based imaging raw data from IMC and MIBI platforms to FR maps are shown Fig 2E and 2F.

## Accurate phenotyping of tissue imaging data across imaging platforms

To further evaluate the performance of our image preprocessing framework, we compared the results of unsupervised clustering after denoising and artifacts correction using two previously published datasets collected by MIBI and Vectra Polaris. The MIBI data are a set of publicly available images from a triple-negative breast cancer cohort (TNBC) with 41 patients [2]. This dataset was analyzed by a computational pipeline [2] that was later wrapped into a user-friendly graphical interface, MAUI [28]. The Vectra dataset includes a subset of 6 patients with ovarian cancer collected in-house and analyzed using inForm [40, 41], a widely-used commercial software package for analyses of multiplex fluorescence imaging data.

We compared the output of our analysis framework with the MAUI pipeline using t-SNE plots comprised of about 200,000 cells from the MIBI breast cancer dataset (Fig 3A and Fig A in S1 Text). The overlaid heatmaps indicate the signal intensities for a subset of non-immune (Keratin6, CD31, SMA, Vimentin) and immune (CD45, CD3, CD4, CD8, CD20, CD68) markers measured from the raw images (top row) or FR maps (middle row). FR maps generated signal intensities ranging between 0 and 1 with improved SNR, as indicated by the more differentiated positive regions in the heat map. The scatter plots (bottom row) demonstrate the non-linear mapping of the average expression values per cell measured from the raw image (x-axis) and the FR maps (y-axis). The Spearman's rank correlation coefficient confirms the presence of a monotonically increasing correlation between the level of expression measured on the raw image compared to the FR map. Additionally, it is noteworthy that as these values increase in the raw image, the values in the FR map level off. Fig B in S1 Text further illustrates this correlation specifically for positive cells of a given marker. We then compared the outcomes of unsupervised clustering (Fig 3B) using normalized marker expression scaled from

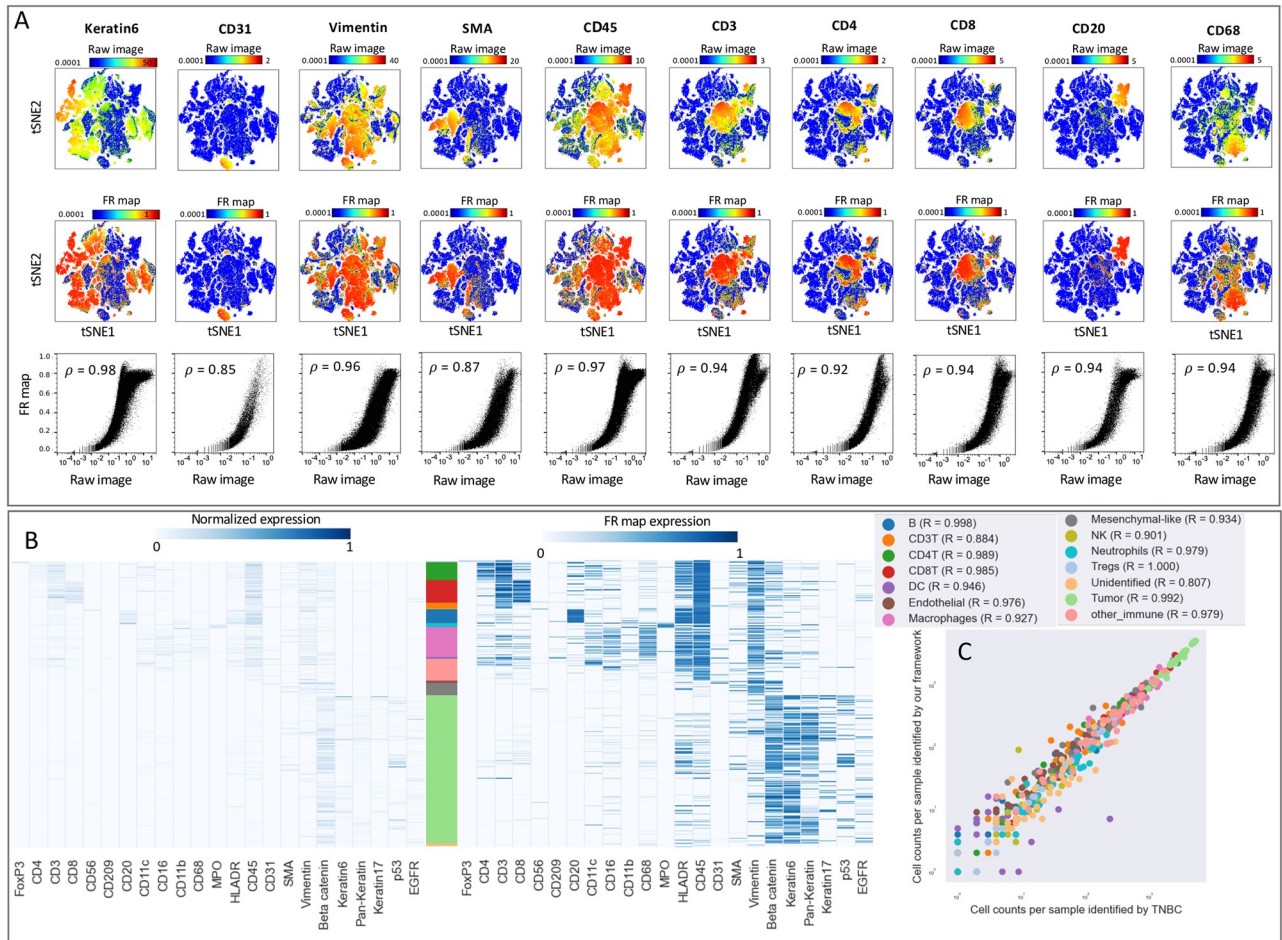

**Fig 3. Our framework delineates cell type compositions in mass ion beam imaging data consistent with MAUI [2]. A**: Marker expression measured per cell using the images (top row) and the FR maps (middle row) are overlaid on the tSNE plot for selected immune and tumor markers. The bottom row demonstrates correlation between marker expression per cell from raw images (x-axis) and FR maps (y-axis). The strength of these correlations are quantified using Spearman's rank correlation coefficient. **B**: Marker expression is shown for cells clustered according to the TNBC study [2] and sorted by cell phenotype. Expression values for each marker are scaled from zero to one (left) or are measured from the FR maps (right). Stacked bar plot shows the abundance of each cell type in the dataset, with corresponding colors specified in the legends of panel C. **C**: Correlation between the frequency of each cell type per patient identified using MAUI in the TNBC study (x-axis) and our proposed framework (y-axis). Pearson coefficient is calculated for each cell type.

zero to one (left) or measured from FR maps (right). To directly compare the two heatmaps and avoid discrepancies introduced by segmentation or clustering, we used the same single-cell segmentation maps and the same cellular phenotypes identified in the TNBC study [2]. Qualitative comparison of the two heatmaps confirms that similar cell populations can be identified using FR maps. For quantitative comparison, we applied the same unsupervised clustering algorithm, FlowSOM [42], to the single-cell information extracted using FR maps. The applicability of our analysis pipline is confirmed by the significant correlation (quantified by Pearson correlation coefficient) in the counts of different cell types identified on a patient-level in the TNBC study and using our framework (Fig 3C). Furthermore, there is a reasonable agreement between the predicted cell types in a comparison of individual cells (Fig A in S1 Text). The discrepancies between the predictions of the two pipelines can be further reduced by manual quality control of the clustering results.

Next, we analyzed ovarian cancer fluorescence imaging data from 6 patients (about 30,000 cells) using inForm software as a benchmark. To avoid any discrepancies caused by differences in the single-cell segmentation, we used the same cell segmentation maps produced by inForm. Fig 4A shows the signal intensities measured from the raw images (top row) or FR maps (middle row) overlaid on tSNE plots. Scatter plots (bottom row) visualize the non-linear mapping of the average expression values per cell measured from the raw image (x-axis) to the FR maps (y-axis) and demonstrate the correlation of signal intensities between the methods. We extracted single-cell information from the FR maps as input to FlowSOM [42] and identified tumor cells and four groups of immune cells. Fig 4B displays a cell-cell comparison of the cell types identified by inForm and our framework. The table entries represent the percentages of cells in the dataset, with the columns listing the identified cell types by the baseline (inForm), and the rows listing the identified cell types by our framework. The pipelines exhibit an

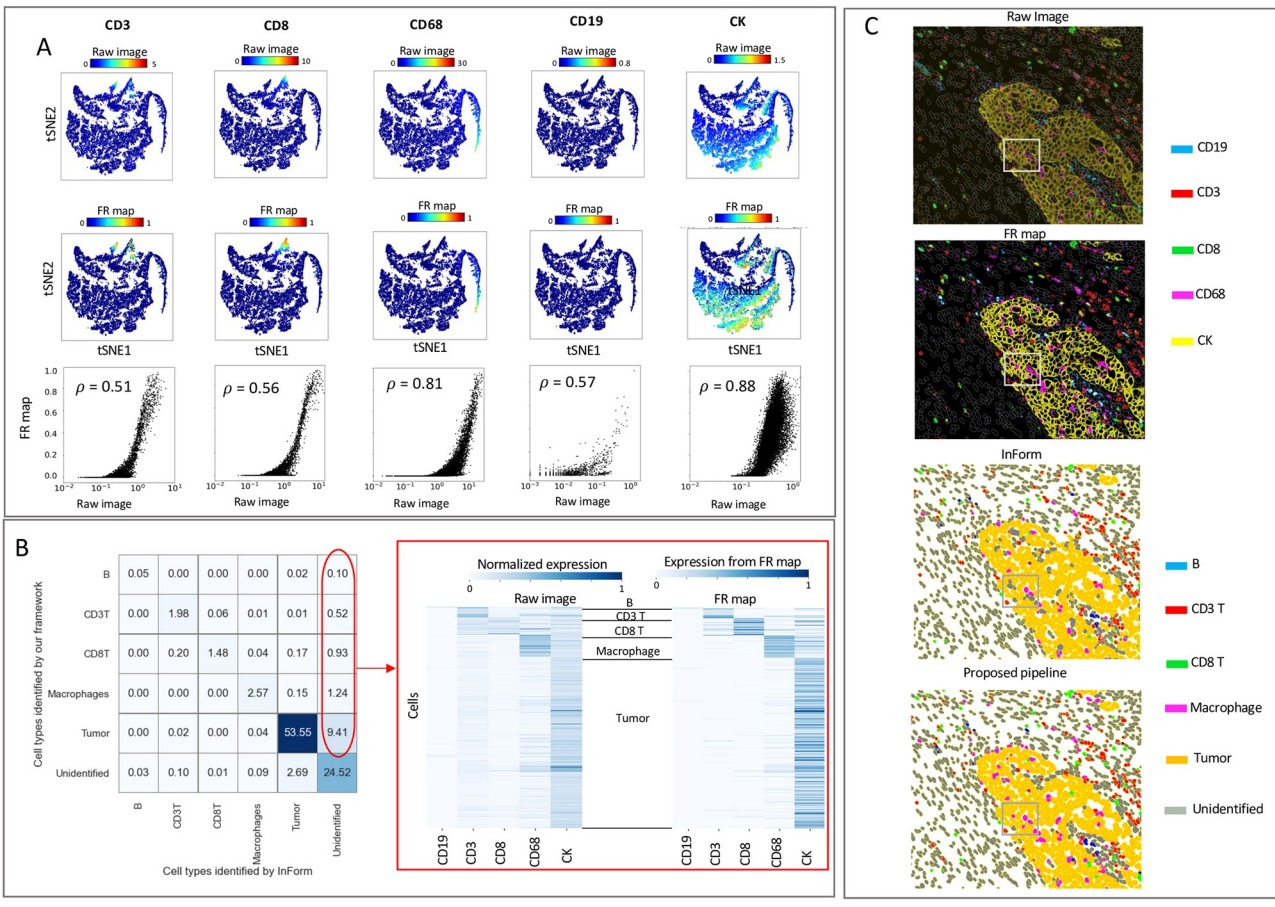

**Fig 4. Our framework delineates cell-type composition in fluorescence imaging data consistent with inForm. A**: Signal intensities for selected immune and tumor markers measured for individual cells using the raw images (top row) or the FR maps (middle row) were overlaid on tSNE plots. Scatter plots (bottom row) demonstrating the correlation between signal intensity per cell from raw images (x-axis) and FR maps (y-axis). The strength of these correlations are quantified using Spearman's rank correlation coefficient. **B**: Cell-cell comparison between the cell type identified by inForm (x-axis) and our framework (y-axis). Table entries indicate the percentage of cells in the dataset to compare the identified cell types by the baselines (columns) and our framework (rows). Heatmap of marker expression for the unidentified cluster by inForm (right); the expression level of markers per cell is measured using the raw image scaled between 0 and 1 (left) or measured from FR maps (right). In both heatmaps the expression level is computed as the summation of pixel values within the boundary of a cell divided by the total number of pixels comprising that cell. **C**: Color overlay of markers CD19, CD3, CD8, CD68, and CK (top); plots compare the stain from the raw image with the corresponding FR map (top). Pseudo-coloring of cell populations compares the predicted cell types by inForm with our framework (bottom).

agreement of approximately 85% (sum of diagonal entries) in identifying cellular phenotypes. Around 12% of the cells remain unidentified by inForm but are assigned to a cell type by our pipeline, which is indicated in red. Many of the cells in inForm's unidentified cluster (Fig 4B-right) have measurable signal intensities that are more clearly defined in the measured counts from the FR map (Fig C in S1 Text). Therefore, our pipeline was better able to discern the phenotypes of these unidentified cells with enhanced SNR. To further investigate these discrepancies, we generated a color overlay of markers from the raw image and the corresponding FR map (Fig 4C-top). Comparing the pseudo-coloring of cell populations predicted by inForm and our pipeline (Fig 4C-bottom) with the protein expression on top, we can easily spot many CK+ tumor cells that are unassigned by inForm.

## Discussion

A goal of image quantification pipelines is to remove noise and artifacts and normalize signal intensities to eliminate batch variations so that imaging data can be combined. The amount of manual curation, dependency on user-defined thresholding parameters, and lack of a unified solution that works across different imaging platforms slows analyses and hinders robust and reproducible results. We developed a framework that overcomes these challenges by replacing the multi-step low-level image processing with a single-step pixel classification. We classify each image such that all categories of undesired signal are placed in a different class from the desired marker signal and continue the downstream analysis using the generated feature representation map of the marker signal. This pipeline opens a new window for integrating data across multiplex imaging platforms and constructing and training generalizable cell-type annotators that can be used in a clinical study with patient and experimental variation.

Using FR maps instead of the raw images has several advantages beyond noise and artifact removal. First, the positive signal intensity values are mapped to normalized values between 0 and 1 in the FR map. Therefore, the artifact removal and normalization are combined into a single step and the normalization is completed at the image level in a very early stage of the analysis.

In addition to noise and artifacts, the platform-specific properties of images such as differences in signal-to-noise ratio, autofluorescence, and background staining are removed from the data. These platform-specific features do not carry useful information and only play confounding roles in downstream analysis. For instance, platform-specific features prevent single-cell segmentation models trained using MIBI data from generalizing on fluorescence-based imaging data [43]. Using our method, discretized and low-SNR signal of MIBI data is converted to continuous and high-SNR signal using FR maps, making the images more similar to fluorescence-based imaging data and making it easier to identify phenotypes. The general idea of pixel classification works across all imaging platforms and thus provides a unified solution for all the preprocessing required for robust cell clustering. Our platform-independent solution eliminates the need for alternative approaches that would otherwise be chosen according to which works best for each imaging technology. For instance, while fluorescence-based imaging data can be denoised using intensity-based methods, MIBI data require density-based methods for denoising.

Previous computational methods for denoising and artifact removal require substantial manual curation and parameter tuning and there is little consensus about which denoising pipeline produces high quality and reproducible results [44]. Many parameters of the algorithms used in the existing pipelines were tuned by hand and evaluated visually, leveraging the expert knowledge of pathologists and biologists. In addition, based on the level and composition of noise, custom-built approaches were optimized by the investigators for different

imaging data [44]. As a result, those methods are prone to human bias and may lack reproducibility. While our proposed method requires expert knowledge for labeling the training data, it offers a unified solution for denoising MIBI data. This is achieved by redefining the problem as a pixel classification problem, where the final value of a pixel is determined by an algorithm that incorporates various imaging features extracted from the raw data. Unlike thresholding parameters that are inferred by the user, our approach eliminates the need for manual thresholding and instead relies on the algorithm's decision-making process to determine the pixel values. Evaluation of functional markers can still be performed with manual thresholding of expression levels, however this process is easier and more robust using FR maps compared to thresholding the raw signal data.

Although advantageous in many ways, supervised pixel level classification is computationally expensive. For instance, generating FR maps for a 41-page image of $800 \times 800 \ \mu m$ ($2048 \times 2048$ pixels) of a highly-expressed marker (e.g. tumor cell markers) in the TNBC data took 140 minutes using an Intel Core i9 CPU-3.30GHz with 128 GB RAM. This time was measured to be less than 90 minutes for low-count markers (e.g. NK cell marker). The time required for labeling the training data should also be considered as 10–15 minutes for adding 20–100 annotations for positive and negative classes depending on the variability in different channels of a marker in a data set. For instance for labeling a marker like Foxp3, 10 annotations per class is sufficient for the classifier to remove the noise from the data. For more ubiquitous signals such as Beta-tubulin or Vimentin more examples are needed. While the run time of our proposed framework is relatively high compared to conventional image preprocessing pipelines, the significantly enhanced SNR reduces manual efforts required downstream for quality control.

## Methods

### Ovarian tumor sample preparation

Images from 12 previously described ovarian tumors were analyzed [41, 45]. Tumors were formalin-fixed, paraffin embedded (FFPE) and assembled in a tissue microarray (TMA) consisting of 2 mm cores Akoya Polaris instrument. TMA construction was approved by Colorado Multiple Institutional Review Board (#17–7788).

### Computational framework

*Data preparation*: Multiplex imaging platforms commonly generate multi-page TIFF or OME TIFF as their primary output. Every page represents a region of interest recorded with a distinct optical or mass channel. For the purpose of pixel classification with our pipeline, a representative training set consisting of multi-page TIFF should be prepared. Every training set TIFF contains the information from a single optical or mass channel, but with representative regions of interest taken from the dataset. To generate a representative training set we use quantitative scores obtained from the CU-IScore scoring system. CU-IScore [46] is a scoring system developed in-house and publicly available, based on the IHC-Profiler method for quantitative evaluation and automated scoring of immunohistochemistry (IHC) images [47]. The CU-IScore algorithm assigns a score from 0–300 based on the pixel intensity and abundance, with higher scores indicating a higher SNR. By utilizing these quantitative scores, we ensure that our training stacks include images with a range of scores, creating a representative subset of the entire dataset for a given marker.

*Pixel classifier training*: Pixel classification has been extensively researched with versatile, interactive, and user-friendly tools available for the analysis of microscopic imaging data [37–39, 48, 49]. Therefore, we did not develop any tool for pixel classification and instead used

TWS [37], a series of library methods that combines Fiji [49] and WEKA [50] and provides an interactive tool for extracting non-linear features and statistical properties of imaging data from user-provided examples. To use the TWS, two sets of inputs are needed from the user: a) multipage TIFF training sets that includes both examples or labels from class I (signal) and class II (noise and artifacts) for each channel of interest, and b) the manual selection of nonlinear features that should be used for classification: To generate label data for the pixel classification training, we provide annotations on the images of the training stack using the TWS interface. Notably, these annotations can be conveniently generated in a single step, encompassing positive signals categorized in class I, as well as all types of noise and artifacts classified within class II. A list of nonlinear features, their parameters, and information gain score that quantifies the importance of each feature in our classification as well as a visualization of the 5 top features are given, respectively, in Tables A and B and Fig D in S1 Text.

*Pixel classifier evaluation*: TWS provides library functions to evaluate the effectiveness of the extracted features in classification as well as the accuracy of used classification algorithms. When using the Random Forest (RF) algorithm for pixel classifcation, we maintain the out-of-bag (OOB) error as a performance measure below a certain threshold. Each decision tree within the RF algorithm is then trained using only a subset of the labeled data, with some samples held out from each tree's training process. The OOB error provides an estimate of the model's performance on unseen data. Additionally, we evaluate the model performance using multiple performance measures including areas under the receiver operating characteristic and the precision-recall curves to explore the sensitivity and specificity of the trained classifier. These built-in performance measures in WEKA package can guide the user in building an appropriate classifier.

*Pixel classification of our datasets*: Since the size of the image sets in both datasets we benchmarked is fairly limited, we did not generate a training subset and instead built a stack from all available regions of interest of a given channel. We note that this approach is only feasible when the dataset size is not very large. Furthermore, we maintained the out-of-bag error below a certain value (around 1%) during training with a Random Forest algorithm with 1000 decision trees. An additional visual inspection of the resulting FR maps was used to correct the pixel labels until optimal results were obtained.

We have developed a series of code that generates the required data structure for the classifier and integrates the output of the classifier with the single-cell segmentation results. Using the the FR maps and the cell segmentation maps, we generate a cell table with cells in row and summary statistics of expression as well as spatial information and morphological properties of cells in columns.

## Data analysis

*Ovarian tumor data collected by Vectra*: using inForm software (Akoya Biosciences), the images were spectrally unmixed, individual cells were identified using DAPI+ nuclei, and the phenotyping algorithms were trained by marking over 100 cells as positive or negative for each of the phenotypic markers in the panel (CD3, CD8, CD68, CD19, and CK). The algorithms were applied to the entire dataset and the data were merged and consolidated in Phenoptr Reports, an open access software package by Akoya Biosciences. Using our proposed pipeline, we generated five stacks from the phenotypic markers where each stack contained 12 images. We then used the TWS tool [37] to train a pixel classifier for each stack and produced a set of FR maps for each marker. We selected Gaussian blur, Sobel filter, mean, median, and entropy kernels for noise reduction and texture filtering. Finally, we extracted single-cell expression values by measuring the signal counts on the FR maps of the positive class. To avoid any

discrepancies caused by differences in the single-cell segmentation, we used the same cell segmentation maps produced by inForm. We used the extracted single-cell information as input to FlowSOM [42] and identified tumor cells and four groups of immune cells. The unidentified cluster includes cells with no positive signal for the present markers or cells with co-expression of mutually exclusive markers.

*Breast cancer tissue collected by MIBI*: We trained a Random Forest classifier using the following features: Gaussian blur, for denoising, mean, median, and entropy kernels for texture filtering, and Sobel filter and difference of Gaussians as edge detectors. We used the single-cell segmentation maps that were generated in TNBC study [2] to extract the single-cell information from our FR maps. Then we followed the exact hierarchical clustering scheme that was originally used to identify the cell types. In doing so, initially FlowSOM was used to cluster the cells into "Immune" and "Non-immune" using 16 markers (CD45, FoxP3, CD4, CD8, CD3, CD20, CD16, CD68, MPO, HLA-DR, Pan-Keratin, Keratin17, Keratin6, p53, Beta catenin, EGFR). Then, using 8 markers (Vimentin, SMA, CD31, Beta-catenin, EGFR, Keratin 17, Keratin 6, Pan-keratin) non-immune cells were clustered to into Epithelial, Mesenchymal, Endothelial and Unidentified. Immune cells were clustered into 12 groups using 13 markers (CD4, CD16, CD56, CD209, CD11c, CD68, CD8, CD3, CD20, HLA-DR, CD11b, MPO and FoxP3). We then manually labeled merged and labeled the clusters with different cell types.

## Supporting information

**S1 Text. Fig A. Top**: Marker expression per cell measured using the images (top row) and the FR maps (middle row) overlaid on the tSNE plot for rest of the immune and tumor markers. The bottom row demonstrate correlation between marker expression per cell from raw images (x-axis) and FR maps (y-axis). The strength and direction of these correlations are quantified using Spearman's rank correlation coefficient. Left: Cells are sorted by cell types identified by our clustering (y-axis) against marker expression (x-axis). Expression values for each marker are measured from the FR maps. Stacked bar plot shows the abundance of each cell type in the dataset. Right: Cell-cell comparison between the cell type identified by the TNBC study versus our framework (left panel). Numbers in table cells indicate the percentage of cells in the dataset where columns and rows, respectively, compare their identified types by the baseline and our framework. **Fig B. Quantifying the correlation between the raw image pixel intensity and FR maps for marker-positive cells**. Correlation plots illustrating marker expression per cell from raw images (x-axis) against FR maps (y-axis) are presented in rows 1 and 3. Specifically, these plots focus on cells where the average FR map values per cell exceed a selected threshold value. To rationalize the chosen threshold values (indicated by red lines), histograms displaying average values per cell measured from FR maps are included, delineating positive cells for a given marker from the negative ones. We note that the mapping from pixel values in the raw image to the FR map is influenced not only by pixel intensity but also by the spatial information of surrounding pixels. Consequently, positive signals may yield large values in the FR map; however, as these values increase in the raw image, the values level off in the FR map. This characteristic does not present any issues, as our framework is not designed to assess the level of expression for functional markers, but rather to determine whether a cell is positive or negative for a given marker. **Fig C**. Cell types clustered by marker expression. Expression values for each marker are measured from the raw image (left) and the FR maps (right). **Table A**. Classification features extracted for the breast cancer and the ovarian cancer datasets. **Table B**. Classification features from Table A in S1 Text in descending order of importance for CD20 marker from breast cancer dataset. **Fig D**. The figure displays the top 5 features utilized for

pixel classification of images from the CD20 channel from the breast cancer dataset.
(PDF)

## Acknowledgments

Technical expertise was provided by the Human Immune Monitoring Shared Resource
(RRID:SCR021985) within the University of Colorado Human Immunology and Immuno-
therapy Initiative and the University of Colorado Cancer Center.

## Author Contributions

**Conceptualization:** Mansooreh Ahmadian, Christian Rickert, Kimberly R. Jordan.

**Data curation:** Mansooreh Ahmadian, Julia Wrobel, Benjamin G. Bitler, Michael Angelo,
Kimberly R. Jordan.

**Formal analysis:** Mansooreh Ahmadian.

**Funding acquisition:** Benjamin G. Bitler, Elena W. Y. Hsieh, Debashis Ghosh, Kimberly R.
Jordan.

**Investigation:** Mansooreh Ahmadian, Julia Wrobel, Benjamin G. Bitler, Fuyong Xing, Elena
W. Y. Hsieh, Debashis Ghosh, Kimberly R. Jordan.

**Methodology:** Mansooreh Ahmadian, Christian Rickert, Fuyong Xing, Debashis Ghosh, Kim-
berly R. Jordan.

**Project administration:** Elena W. Y. Hsieh, Debashis Ghosh, Kimberly R. Jordan.

**Resources:** Mansooreh Ahmadian, Christian Rickert, Angela Minic, Benjamin G. Bitler,
Michael Angelo, Kimberly R. Jordan.

**Software:** Mansooreh Ahmadian, Julia Wrobel.

**Supervision:** Fuyong Xing, Elena W. Y. Hsieh, Debashis Ghosh, Kimberly R. Jordan.

**Validation:** Mansooreh Ahmadian, Debashis Ghosh, Kimberly R. Jordan.

**Visualization:** Mansooreh Ahmadian, Kimberly R. Jordan.

**Writing – original draft:** Mansooreh Ahmadian, Kimberly R. Jordan.

**Writing – review & editing:** Mansooreh Ahmadian, Christian Rickert, Angela Minic, Julia
Wrobel, Benjamin G. Bitler, Fuyong Xing, Michael Angelo, Elena W. Y. Hsieh, Debashis
Ghosh, Kimberly R. Jordan.

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
