## [Decision Letter · Decision Letter 0]

29 Mar 2023

Dear Dr. Jordan,

Thank you very much for submitting your manuscript "A Platform-Independent Framework for Phenotyping of Multiplex Tissue Imaging Data" for consideration at PLOS Computational Biology.

As with all papers reviewed by the journal, your manuscript was reviewed by members of the editorial board and by several independent reviewers. In light of the reviews (below this email), we would like to invite the resubmission of a significantly-revised version that takes into account the reviewers' comments.

In addition to addressing all reviewers' comments, please ensure that details on training, *along with training data* are made publicly available either on the project's GitHub repository or on another data sharing platform of your choice (eg Zenodo) before submitting the revised version.

We cannot make any decision about publication until we have seen the revised manuscript and your response to the reviewers' comments. Your revised manuscript is also likely to be sent to reviewers for further evaluation.

Sincerely,

Virginie Uhlmann

Academic Editor

PLOS Computational Biology

Lucy Houghton

Staff

PLOS Computational Biology

In addition to addressing all reviewers' comments, please ensure that details on training, *along with training data* are made publicly available either on the project's GitHub repository or on another data sharing platform, (eg Zenodo) before submitting the revised version.

Reviewer's Responses to Questions

**Comments to the Authors:**

Reviewer #1: an individual file has been uploaded.

Reviewer #2: The authors make a strong case that pixel classification is a good approach to combine several steps of noise and artifact removal into one step. They demonstrate how applying this to multiplex tissue images can provide pixel values that are good for downstream analysis.

This is an important result and will give confidence to other researchers to follow this approach, thereby enabling the possibility of having a significant impact on the field of multiplexed image processing.

While this idea may occur to other people, the paper provides a useful contribution in showing that this is an effective technique by providing detail of the transformation as well as showing that it performs well in real biological analysis problems.

The authors make a clear case as they state "The pixel density should be considered along with the intensity." and show this point throughout the paper.

The authors also talk about how their technique performs normalization, and I have some concerns about this, and give more details in major issue 1.

My overall impression is that it is a quality, clear and useful paper, though I do have a major concern that should be addressed before publication.

Major issue:

1. One major issue is how the authors talk about the normalization their algorithm performs, and I am concerned about its relationship to quantitative downstream analysis.

One of the main motivating factors in this work is to provide normalization. This is evidenced in the abstract where 'normalized pixel values' are mentioned as well as on page 2: "Therefore, high-quality imaging data should ideally satisfy the following criteria: 1) the data should exhibit high signal-to-noise ratios (SNR) and be as free as possible from artifacts; and 2) be appropriately normalized to remove non-biological signal variability within and across acquisition batches and tissues."

and also on page 2:

"Our approach combines denoising and removal of various artifacts into a single pixel classification step, outputs a feature representation map (FR map) that eliminates the need for any further normalization process"

Normalization often refers to a process where the mean or maximum value of some pixels are shifted, but that preserves the relationship between pixels such that pixels that are brighter in the original image are brighter in the processed image. Normalization can also be an appropriate way to refer to batch correction as the authors do on page 2: "be appropriately normalized to remove non-biological signal variability within and across acquisition batches and tissues". This statement seem correctly supported by the work, as the normalization they perform does seem to occur over batches and tissues, as is desirable.

However, from what I can see, the 'normalization' in this work also 'normalizes' between different cells within one image, so pixels that are part of class 1 'signal' are not normalized in a way that is desirable for most downstream analysis. Within one image, if pixel A and pixel B are part of the signal class and pixel A is brighter than pixel B in the original image, then it is possible than pixel B becomes brighter than pixel A in the processed image. To put this in more mathematical terms, the mapping within the signal class of pixels in not monotonically increasing and this distortion of pixel values in the final image could provide problems in any cell-level downstream analysis.

One example of this distortion of class I pixels can be seen in Figure 2d. Consider the cell in the bottom middle of the image, which I will call cell D (location is horizontally between B and C, and vertically much below both of them). If you compare the brightness cell D compared to A before and after: before cell D is much darker than cell A, and afterwards cell D and cell A look about equal brightness. This loss of distinction could negatively impact quantitative analysis of the signal.

Another way to see this is in the graphs in figure 3a. For example, in the CD3 case, some pixels that raw image value of about 0.5 map to the FR map with a value of 1, but some pixels with a raw image value of 2 have an FR map value of 0.8, inverting the brightness of the original values. This might be pixels from different images, but it is not possible to tell from the data presented and so there remains a concern about how the normalization modifies values within the signal class within one image.

I can think of two ways to address major issue 1.

1. A graph of pixel intensities of the pixel in class 1 (or above 0.5 in class 1). This graph would be similar to the graphs in Figure 3a, but should be split by different origin image so that it is possible to see whether higher signal pixels in the original image produce higher signal pixels in the processed image. It may be helpful if this graph was not log scale. If a graph like this show that signal is mostly monotonically increasing within each image, then that would remove this concern.

2. Description of some of this issue in the discussion, highlighting the non-linearity within the signal that is created, acting as a warning to those who would do downstream analysis on the FR map.

Minor issues:

1. A related issue within this is that the paper does not seem to describe what exactly the feature representation quantifies. It may be the probability of being within a class, or that probability multiplied by original signal, or something more complex. An explanation or a link to where this is explained is essential for helping to deal with this issue.

2. In the abstract 'pixel-accurate representations' could be changed to 'pixel-resolved representations' or 'pixel-resolution representations' or 'pixel-scale representations' or 'pixel-precise representations' or something similar to make it clearer that you are not talking about the accuracy of pixel values, but are making a scale/size claim.

3. In the introduction you mention (page 2) "user-defined thresholding parameters that are prone to bias". This is fine, but it may be worth adding a discussion point about how your method needs training data, and that training data can be prone to some bias too.

4. On figure 3, the FR map y scale values should be quote as '0.8' not '0.800'

5. On the issue of uniqueness, it would be worth describing how your work is unique against Giannakeas et al. "Segmentation of microarray images using pixel classification—Comparison with clustering-based methods". This could include just having two classes, using multiplexed images, or other things.

Other points:

1. The introduction is extremely clear and well-motivated, particularly the first paragraph.

2. The document is also filled with very helpful summary statements including:

"For instance, while fluorescence-based imaging data can be denoised using intensity-based methods, MIBI data require density-based methods for denoising.". This statement helps explain why the proposed method is suitable across multiplexed imaging types.

3. The authors state that data is available on request. This is acceptable in my opinion, though the authors may want to consider making the data publically available.

I am available to look at any revised version.

Reviewer #3: This manuscripts makes an attempt at establishing a sought after “framework” or “pipeline” for the analysis of multiplex fluorescence and use it for cell type discovery.

However it falls onto a rebranding problem. This work is not platform independent it uses a series of modules that already exist and are in use and depend on the platforms they are installed on.

The authors often mention the many existing technologies around mIF but compare to very few and a closed one, inform. I suggest reading the review by Parra et al [1] it might also be good to reduce unnecessary references.

The authors claim that they have found a way to deal with intensity normalization issues by segmenting (which they rebrand as “FR maps”). However the segmentation is done with the well known Ilastik whose features do not include intensity insensitive features, making intensity normalization still a problem. With the segmentation there is still a threshold (or several, in a tree) to be made amongst features that are sensitive to intensity changes. In Solorzano et al [2] an attempt is done to overcome this problem with the introduction of an intensity insensitive feature and the introduction of deep learning features, even a segmentation is also done. In [3] the QuPath software is used and it combines what is done here (in a single and truly platform independent framework with a friendly user interface) by extracting features like Ilastik (plus deep learning features) and using Random Forests and also Deep learning to do segmentation. I am aware that there is a difference between the amount of channels available in the data used in [3] and this manuscript but the platform works for both. [2] also compares with CODEX data. Why not compare your methodology with your references 2 and 4 (Leet and Goltsev)?

Comparing with InForm is not very informative unless you happen to own an Akoya device, but I do like the comparison with MAUI. I would still like to see a comparison with QuPath. In general, classical image features have been shown to not be enough to capture variety and deep learning features should be explored, how to do it is another story, it has to be chosen by the researcher, all platforms, theoretical and software already exist, so in this aspect this manuscript doesn’t bring anything new. Additionally Weka is regarded in the bioinformatics scene as a more exploratory tool but not useful for scaling to bigger experiments. Speaking of which, there is no comparison in terms of time and performance, any time I use InForm it is extremely slow, although it lets you do some parameter modifications, and parameters are also not shared or discussed. How are MAUI and InForm different/similar?

When Random Forests are used, it is common to show feature importance, this way one can actually observe which features were more meaningful for the classification so this would be interesting to see.

The overall idea it’s still good, as a showcase of the use of simple, common and well known and used methodologies and stresses the importance of denoising and noise modeling and discusses the issue of normalization. So in summary the strength of this manuscript lies not in a novel or platform independent framework/pipeline but as a showcase of a commonly used set of methodologies and showing that it works. Data like the one used has already several studies so I am sure there are more instances to compare to.

Figure S1 does worry me a bit as there is what seems to be a big confusion with Macrophages and Tumor, which is actually common in MIBI and mIF data. While the plot looks nice, numbers would be more appreciated and easy to compare to.

A final comment is that at the time I finished this review, the repository’s readme doesn’t really explain what is contained in the rest of the code and was last updated 5 months ago with no modifications (even if it says “repo under development” there has been no development. At least the code has comments which is good. Having to separate the images into channels out side the code makes it an additional step that contributes to this work not truly being platform independent, there are many steps and requirements that are not really discussed.

[1] Parra ER, Francisco-Cruz A, Wistuba II. State-of-the-Art of Profiling Immune Contexture in the Era of Multiplexed Staining and Digital Analysis to Study Paraffin Tumor Tissues. Cancers (Basel). 2019 Feb 20;11(2):247. doi: 10.3390/cancers11020247. PMID: 30791580; PMCID: PMC6406364.

[2] Solorzano L, Wik L, Olsson Bontell T, Wang Y, Klemm AH, Öfverstedt J, Jakola AS, Östman A, Wählby C. Machine learning for cell classification and neighborhood analysis in glioma tissue. Cytometry A. 2021 Dec;99(12):1176-1186. doi: 10.1002/cyto.a.24467. Epub 2021 Jun 22. PMID: 34089228.

[3] Viratham Pulsawatdi A, Craig SG, Bingham V, McCombe K, Humphries MP, Senevirathne S, Richman SD, Quirke P, Campo L, Domingo E, Maughan TS, James JA, Salto-Tellez M. A robust multiplex immunofluorescence and digital pathology workflow for the characterisation of the tumour immune microenvironment. Mol Oncol. 2020 Oct;14(10):2384-2402. doi: 10.1002/1878-0261.12764. Epub 2020 Sep 1. PMID: 32671911; PMCID: PMC7530793.

**Have the authors made all data and (if applicable) computational code underlying the findings in their manuscript fully available?**

Reviewer #1: **No: **code to run their processing is provided on github. However, the training data for this dataset is only available upon request.

Reviewer #2: **No: **Authors say that some of the data is available on request, so I am not sure whether this is acceptable to you.

Reviewer #3: Yes

PLOS authors have the option to publish the peer review history of their article (what does this mean?). If published, this will include your full peer review and any attached files.

Reviewer #1: No

Reviewer #2: **Yes: **Benjamin Woodhams

Reviewer #3: No
---

## [Decision Letter · Decision Letter 1]

9 Jul 2023

Dear Dr. Jordan,

Thank you very much for submitting your manuscript "A Platform-Independent Framework for Phenotyping of Multiplex Tissue Imaging Data" for consideration at PLOS Computational Biology. As with all papers reviewed by the journal, your manuscript was reviewed by members of the editorial board and by several independent reviewers. The reviewers appreciated the attention to an important topic. Based on the reviews, we are likely to accept this manuscript for publication, providing that you modify the manuscript according to the review recommendations.

Please address the comments of Reviewer 1 and add a paragraph discussing the high FR map signal/original pixel intensity relationship and how that may limit the proposed method as suggested by Reviewer 2.

Sincerely,

Virginie Uhlmann

Academic Editor

PLOS Computational Biology

Lucy Houghton

%CORR_ED_EDITOR_ROLE%

PLOS Computational Biology

Please address the comments of Reviewer 1 and add a paragraph discussing the high FR map signal/original pixel intensity relationship and how that may limit the proposed method as suggested by Reviewer 2.

Reviewer's Responses to Questions

**Comments to the Authors:**

Reviewer #1: I thank the authors for the revision and feel that my comments have been sufficiently addressed. Two small comments remain which should be easy to address and the authors should also cite the meanwhile not only on BioRxiv available paper from the Thorek lab (https://doi.org/10.1038/s41467-023-37123-6).

Regarding major comment 2. Fig. 2A makes for better visual impression of cross-talk albeit unfortunately now the histogram for that one single CD20 positive cells is hard to appreciate. Potentially a zoom in could help.

Additionally I was not aware of the dual labelling process for noise first and then artefacts. Might be worth to specify this in the methods if the authors believe that this is superior over a one step labelling (noise+artefacts). Not fully clear to me what the difference would be.

Regarding minor comment 1: unclear what the difference between Fig. 3b right side and Fig. S1 is. The heatmaps look pretty much identical.

Reviewer #2: Major issue 1 - partially addressed

I thank the authors for their replies and clarifications on this issue. Firstly the addition of spearman's rank is a helpful addition. Secondly, my concerns with Figure 2d have been successfully addressed. Thirdly, including further discussion about non-linearity has been helpful. I would like to clarify my point since it may not have been clear in my original comment and some of my concern remains.

I remain concerned about the correlation of the data within the upper values of the FR map, which I believe are the most relevant values for most applications.

Many downstream analyses will focus on quantifying signal that is found at higher values of FR map (this is also higher values of original signal intensity in general). One example of how this could happen is by choosing to analyze data that is only within cells to compare cells to each other. An example of this in the paper is where it says in the Figure 1 caption "Marker expression within the border of each cell is then measured from the class I FR maps." By doing an analysis like this, the lower values of FR map become irrelevant and the overall monotonic increase is less relevant. In this scenario, the most relevant question is: is the FR map signal within the upper values monotonically increasing with original pixel intensity?

While the authors have very helpfully described this monotonic increase for the whole dataset and helpfully quantified the correlation over the whole thing, it is still a matter of concern about how well the FR map values reflect the original pixel intensity for areas of high FR map that are likely to be analyzed as part of signal. Part of my motivation for considering this is that in my mind, expression of signal certainty is not quite the same as increased signal.

This paper shows this approach works overall. However, I am trying to make this limitation, if it exists, more clear. There are a few options I can think of to resolve this problem.

Option 1: Add a written caveat (perhaps in the discussion) that monotonic increases are less clear for high FR values. (or something like you might lose dynamic range up at those values)

Option 2: Add a supplementary figure that replicates figure 3a (row 3) but plotted on a linear scale, not on a log scale, for FR map values only above 0.5 (spearman's rank optional), just to make this area of the graph clearer (it may be that the current graph is clear enough in the editor's opinion)

Option 3: A supplementary table of spearman's rank for FR map >0.5 for the data is figure 3 row 3

It may be that the editor thinks there is already enough information about correlation at high FR values in the manuscript. In which case, that is fine.

Minor issue 1 - fully addressed

Minor issue 2 - fully addressed

Minor issue 3 - fully addressed

Minor issue 4 - fully addressed

Minor issue 5 - fully addressed

Other points 1 - fully addressed

Other points 2 - fully addressed

**Have the authors made all data and (if applicable) computational code underlying the findings in their manuscript fully available?**

Reviewer #1: None

Reviewer #2: Yes

PLOS authors have the option to publish the peer review history of their article (what does this mean?). If published, this will include your full peer review and any attached files.

Reviewer #1: No

Reviewer #2: **Yes: **Benjamin Woodhams

Figure Files:

Data Requirements:

Reproducibility:

References:

---

## [Editor Report · Decision Letter 2]

14 Aug 2023

Dear Dr. Jordan,

We are pleased to inform you that your manuscript 'A Platform-Independent Framework for Phenotyping of Multiplex Tissue Imaging Data' has been provisionally accepted for publication in PLOS Computational Biology.

Best regards,

Virginie Uhlmann

Academic Editor

PLOS Computational Biology

Lucy Houghton

%CORR_ED_EDITOR_ROLE%

PLOS Computational Biology

---

## [Editor Report · Acceptance letter]

13 Sep 2023

PCOMPBIOL-D-22-01763R2 

A Platform-Independent Framework for Phenotyping of Multiplex Tissue Imaging Data

Dear Dr Jordan,

I am pleased to inform you that your manuscript has been formally accepted for publication in PLOS Computational Biology. Your manuscript is now with our production department and you will be notified of the publication date in due course.

With kind regards,

Judit Kozma
